# Enhanced Anti-Inflammatory Activity of Tilianin Based on the Novel Amorphous Nanocrystals

**DOI:** 10.3390/ph17050654

**Published:** 2024-05-17

**Authors:** Min Sun, Mengran Guo, Zhongshan He, Yaoyao Luo, Xi He, Chuansheng Huang, Yong Yuan, Yunli Zhao, Xiangrong Song, Xinchun Wang

**Affiliations:** 1Department of Pharmacy, First Affiliated Hospital of Shihezi University, Shihezi 832008, China; sunmin1997@hotmail.com (M.S.); chuansheng_huang@hotmail.com (C.H.); yuanyong_1967@hotmail.com (Y.Y.); zhaoyunli2024@hotmail.com (Y.Z.); 2School of Pharmacy, Shihezi University, Shihezi 832008, China; 3Department of Critical Care Medicine, Department of Clinical Pharmacy, State Key Laboratory of Biotherapy and Cancer Center, West China Hospital, Sichuan University, Chengdu 610000, China; gmran_2020@hotmail.com (M.G.); zhongshan_he@hotmail.com (Z.H.); lyy7208570@hotmail.com (Y.L.); xhe@scu.edu.com (X.H.)

**Keywords:** tilianin, nanocrystals, amorphous, macrophage polarization, reactive oxygen species, anti-inflammatory

## Abstract

Tilianin (Til), a flavonoid glycoside, is well-known for its therapeutic promise in treating inflammatory disorders. Its poor water solubility and permeability limit its clinical applicability. In order to overcome these restrictions, an antisolvent precipitation and ultrasonication technique was used to prepare amorphous tilianin nanocrystals (Til NCs). We have adjusted the organic solvents, oil-to-water ratio, stabilizer composition, and ultrasonic power and time by combining single-factor and central composite design (CCD) methodologies. The features of Til NCs were characterized using powder X-ray diffraction (PXRD), scanning calorimetry (DSC), and transmission electron microscopy (TEM). Specifically, the optimized Til NCs were needle-like with a particle size ranging from 90 to 130 nm. PVA (0.3%, *w*/*v*) and TPGS (0.08%, *w*/*v*) stabilized them well. For at least two months, these Til NCs stayed amorphous and showed an impressive stability at 4 °C and 25 °C. Remarkably, Til NCs dissolved almost 20 times faster in simulated intestinal fluid (SIF) than they did in crude Til. In RAW264.7 cells, Til NCs also showed a better cellular absorption as well as safety and protective qualities. Til NCs were shown to drastically lower reactive oxygen species (ROS), TNF-α, IL-1β, and IL-6 in anti-inflammatory experiments, while increasing IL-10 levels and encouraging M1 macrophages to adopt the anti-inflammatory M2 phenotype. Our results highlight the potential of amorphous Til NCs as a viable approach to improve Til’s anti-inflammatory effectiveness, solubility, and dissolving rate.

## 1. Introduction

Til, also known as acacetin-7-glucoside, is an active flavonoid glycoside (Figure 1 and Appendix A) that has been isolated from a number of medicinal plants, including Dracocephalum moldevica, a plant used in Chinese traditional medicine [1,2,3]. Numerous biological activities, such as those that are antidiabetic [4], anti-inflammatory [5], antioxidant [6], and antidepressant [7], have been documented. Til has additionally been demonstrated to exhibit cardioprotective [8,9], neuroprotective [10], and antihypertensive properties.

In the past few decades, the outstanding anti-inflammatory properties of Til have drawn an increasing amount of attention. Many diseases can develop into inflammatory states. Macrophages are the predominant immune cell population. The phenotypic and functional plasticity of macrophages are greatly enhanced by their polarization, and this is directly associated with the onset, progression, and outcome of inflammation in various situations. The primary distinction between M1 and M2 macrophage phenotypes is in their severe polarization, which places them at a pivotal point in the pathophysiology of numerous inflammatory illnesses, including enteropathy, hepatitis, pulmonary disorders, and cardiovascular diseases [11,12,13,14]. Thus, we looked into the possibility that Til NCs could alter macrophage polarization in the current investigation. Specifically, proinflammatory cytokines including TNF-α, IL-1β, and IL-6 are produced and secreted by M1-type macrophages in response to various stimuli, such as lipopolysaccharide (LPS) and IFN-γ [15,16]. The nicotinamide adenine dinucleotide phosphate (NADPH) oxidase complex is functionally activated by M1 macrophages, resulting in the production of reactive oxygen species (ROS) that are linked to tissue damage and pathogen clearance [17]. Concurrently, Th1 and natural killer cells are drawn in by the chemokine receptor ligands that M1-type macrophages express. This results in a prolonged inflammatory response that is necessary for the removal of cellular pathogens [18]. Nonetheless, M2 macrophages’ primary roles include reducing inflammation, eliminating apoptotic cells and cell debris, and taking part in tissue fibrosis and repair [19,20,21]. To counteract the chronic inflammatory response triggered by M1-type macrophages, they produce chemokines and secrete anti-inflammatory molecules including TNF-α and IL-10, which activate functional anti-inflammatory regulatory pathways [22,23]. Thus, one effective way to reduce inflammation is to control the phenotypic change in macrophages and the release of inflammatory chemicals.

Research has demonstrated that Til can raise TNF-α and IL-1β mRNA levels in primary Ldlr^-/-^ mice’s peritoneal macrophages when they are activated by LPS [24]. Furthermore, Til has the potential to stop IκB kinase activation as well as the phosphorylation and degradation of the IκBα protein, which occurs prior to nuclear factor-κB (NF-κB) activation. Til inhibited the LPS-induced macrophage inflammatory response while producing vascular cell adhesion molecule (VSMC) proliferation and migration, as shown by Shen et al.‘s research [25]. Til also reduced TNF-α levels. It was also suggested by them that Til’s anti-inflammatory action on vascular smooth muscle cells and macrophages was mostly brought about via inhibiting the TNF-α/NF-κB pathway. In a different investigation [26], Til was shown to have a strong protective effect against LPS-induced acute lung injury (ALI) in mice by effectively preventing the condition both in vitro and in vivo. Although there is strong evidence that Til may effectively block proinflammatory factors in the treatment of inflammation-related disorders, little is known about how they affect macrophage polarization in inflammatory tissues.

Til is a BCS IV medication, and because of its low water permeability and solubility, it has a low oral bioavailability. Consequently, it is crucial to develop innovative medication delivery techniques to increase therapy efficacy. Til-loaded composite phospholipid liposomes and Til-PLGA block copolymer nanoparticles co-modified with transcription activators and polyethylene glycol are two methods to enhance the oral uptake of Til [27]. Despite the fact that some carriers function well in vivo, they can have drawbacks such low drug loading, unstable materials, and possible toxicity.

In recent years, drug nanocrystals composed of drugs and stabilizers have been considered one of the most promising nanotechnologies, which can significantly improve the dissolution rate and solubility by reducing the particle size to the nanoscale. Nanocrystals have unique advantages, such as high drug content, high bioavailability, few adverse reactions, and strong adhesion ability to biofilms and tissues [28,29]. It is preferable to the development of BCS IV pharmaceutical formulations (e.g., Til) [30]. Antisolvent precipitation is a method of precipitating a compound or substance by controlling the temperature and the ratio of the solvent mixture. The method usually involves dissolving an insoluble compound in an organic solvent and then gradually adding another antisolvent. Under the antisolvent procedure, the insoluble compound will precipitate to form crystals. This method can be used to prepare compounds of high purity, as well as micro-nanoparticles. To control the particle size in this procedure, ultrasonication was subsequently applied to inhibit the crystal growth by the ultrasonic cavitation effect.

This study aims to develop Til nanocrystals (Til NCs) by an antisolvent precipitation and ultrasonication method [31,32,33]. First, the central composite design (CCD) was utilized to optimize the formulation after a thorough investigation of the formulation’s affecting elements was conducted using the single-factor method. Subsequently, the shape and physicochemical properties of Til NCs were explained. More research was conducted on the stability and dissolution behavior in vitro. Lastly, we looked into Til NCs’ cellular absorption, safety, and anti-inflammatory properties.

## 2. Results and Discussion

### 2.1. Formulation Optimization

#### 2.1.1. Selection of Organic Solvents, Stabilizers, and Process Parameters

The formulation screening results of Til NCs are presented in Figure 2. The type of organic solvents, ratio of organic phase to water phase, ultrasonic power, ultrasonic time, and stabilizers were investigated.

The antisolvent precipitation ultrasonic method was used to obtain Til NCs. In the preparation process, supersaturation induced by organic solvents is critical for controlling the nucleation process, especially for poorly water-soluble drugs such as Til. Thus, it is essential to screen an appropriate organic solvent to achieve higher supersaturation and induce quicker nucleation, resulting in an ideal crystal particle size [34]. As revealed in Figure 2A, large particle aggregates were observed when using DMF, different proportions of acetone and DMF, and 70% ethanol. However, the solubility of Til in DMF:ethanol (1:1) was significantly higher than that in other solvents, and DMF:ethanol (1:1) could generate a nanocrystal with a smaller particle size and PDI, which was beneficial to the stability of the preparation. Thus, DMF:ethanol (1:1) was the optimal solvent. The addition of ethanol was supposed to reduce the viscosity of DMF, accelerating the diffusion of the organic solvent and the nucleation rate. In addition, the ratio of the organic phase to the water phase also influences the nucleation of drugs. As shown in Figure 2B, a smaller particle size was obtained when the ratio was 1:20. Ultrasonic power and time are key process parameters that can promote nucleation by producing acoustic cavitation in solution and thus shorten the induction time, resulting in a nanocrystal with a uniform particle size distribution [35]. As shown in Figure 2C,D, with increasing ultrasonic power and time, the particle size and PDI of the Til NCs gradually decreased, but when the ultrasonic power and time were further increased, the particle size of the Til NCs tended to increase. The reason may be that the local generation of heat with increasing ultrasonic power and time will cause particle agglomeration and aging. Therefore, the optimal ultrasonic power was 120 W, and the ultrasonic time was 30 min.

The stabilizer is a critical component for NCs with the benefit of inhibiting crystal nucleation and aggregation. The stabilizers can generally be classified into two categories. One is spatial stabilizers, such as F127, F68, PVP K30, PVA, and other polymer materials with spatial structures. These kinds of stabilizers can inhibit agglomeration and sedimentation for nanocrystals by forming hydrophilic films or hydrogen bonds on the surface of nanocrystals. Another class of stabilizers is ionic stabilizers, such as sodium deoxycholate, TPGS, SDS, and CTAB, which can maintain the stability of the nanocrystals by providing sufficient electrostatic repulsion between particles. The screening results of the types and concentrations of stabilizers are presented in Figure 2E–P. From the results, under identical preparation conditions, the particle size and PDI of Til NCs were effectively reduced by the spatial stabilizer PVP K30 or PVA (Figure 2G,H), while F68 and F127 failed to reduce the particle size and PDI of Til NCs (Figure 2E,F). In addition, the particle size of NCs stabilized by PVP K30 changed greatly when stored at room temperature for 24 h (Figure 2G), indicating that the NCs were unstable. The results of PVA concentration screening showed that the particle size of Til NCs decreased with increasing PVA concentration. Therefore, PVA was chosen as the spatial stabilizer of Til NCs in this study. Furthermore, Figure 2I–L shows the particle sizes of Til NCs stabilized by different types of ion stabilizers. The obtained NCs are all approximately 100 nm. As shown in Figure 2M–P, the zeta potential of Til NCs containing the CTAB stabilizer is negatively charged (Figure 2P), which is not conducive to drug absorption. In contrast to negatively charged NCs, positively charged NCs can facilitate mucus attachment [36] and thereby easily penetrate through the mucus layer. However, sodium deoxycholate is unstable in gastric acid, and SDS produces a large amount of foam during the preparation process, which affects the experimental results. The addition of TPGS reduced the particle size and PDI of the Til NCs. In addition, TPGS can open tight junctions between epithelial cells, which is beneficial for nanoparticles to promote cellular absorption through paracellular pathways. It is also an inhibitor of P-glycoprotein (P-gp) efflux, which can reduce drug efflux and improve the oral bioavailability of drugs [37]. Therefore, we chose PVA and TPGS as the main stabilizers to dominate the formation process of Til NCs. The effects of PVA and TPGS concentrations on the particle size and PDI of Til NCs were investigated (Figure 2H,K). It was demonstrated that with an increase in PVA concentration, the particle size of Til NCs decreased. However, when the PVA concentration is high, the particle size of the nanosuspension does not change much, while the PDI increases. For TPGS, the particle size and PDI of Til NCs tended to increase with increasing TPGS concentration. According to one interpretation of these data, the stabilizer masks the hydrophobic region of the nanoparticle and reduces the amount of free stabilizer in the solution when its concentration is low enough to cover the entire hydrophobic surface. Nevertheless, redundant stabilizers dissolve more medication or form micelles when the concentration of stabilizers rises, which causes particle aging and instability. PVA and TPGS were chosen for CCD improvement in light of this.

#### 2.1.2. Optimization Using the CCD Method

CCD is a statistical method that uses a multiple quadratic regression equation to fit the functional relationship between the factors and the response values and finds the optimal process parameters through the variance analysis of the mathematical model. In this study, a three-factor, three-level trial was employed, and 17 baths were taken. Table 1 lists the trial’s setup and outcomes. To lessen systematic errors brought on by outside influences, experiments were conducted at random. Using the answer and associated data from the 17 trials, Design Expert 12 software was utilized to fit linear, two-factor (2F), and quadratic models concurrently. As a result, the ideal formula for Til NCs was determined and predicted using the quadratic equation. The quadratic model’s polynomial regression equation is as follows:Y = 104.38 − 2.40X_1_ − 1.60X_2_ − 2.00X_3_ + 0.9250X_1_X_2_ + 6.27X_1_X_3_ + 3.23X_2_X_3_ − 1.78X_1_^2^ − 8.78X_2_^2^ + 0.0725X_2_^3^ (R^2^ = 0.9264, *p* < 0.01)

The primary parameters influencing the particle size of Til NCs were determined by an analysis of variance (analysis of variance (ANOVA), Table 2): ultrasonic power (X_1_), PVA concentration (X_2_), and TPGS concentration (X_3_). In the ANOVA, F_Y_ = 9.79, which meant that the model was meaningful, while *p* = 0.0033, indicating that the response-surface model was very significant. The loss of fit term *p* = 0.1912 indicated that the model fit degree was good and the model was established. The correlation coefficient R^2^ was 0.9264, and the adjustment coefficient R^2^_adj_ was 0.8317, indicating that the actual value and the predicted value had a good degree of fit. The test error was small, indicating that the fitted regression model could be used for the analysis and prediction of the optimal prescription parameters of Til NCs with certain values and reliability. The F value in Table 1 can judge the degree of influence of the respective variables on the particle size. The larger the F value is, the more significant the influence of the independent variable on the particle size. From Table 2, F_X1_ = 6.27, F_X2_ = 2.78, and F_X3_ = 4.35, indicating that the degree of influence of each factor on particle size was X_1_ > X_3_ > X_2_. The effect of the quadratic factor X_2_^2^ on particle size was extremely significant (*p* < 0.01).

When one of the factors is fixed, the optimal values of the other two factors selected to appear under interactive variation are all around the initially selected zero points, which proves that the initial conditions for optimization are better in the single-factor experiment. The interaction effects of various factors on Til NCs particle size are presented in Figure 3. The steepness of the response-surface map reflects the interaction between each factor and the response value. The steeper the slope of the surface plot is, the more significant the effect of the interaction between the two factors on the response value, and the smoother the slope is, the less significant the interaction between the two factors. The more oval or saddle shape of the contour plot indicates that the interaction between the two factors is significant, and the more circular shape indicates that the interaction between the two factors is not significant. In Figure 3A,B, the particle size of Til NCs showed a decreasing trend with increasing ultrasonic power. In Figure 3B,C, a higher TPGS concentration resulted in smaller particles for Til NCs. In Figure 3A,C, the particle size increased with increasing PVA concentration, while when the concentration was over 0.2% (*w*/*v*), the particle size decreased. The interaction degree of each influencing factor on the particle size was X_1_X_3_ > X_2_X_3_ > X_1_X_2_, and the interaction between X_3_ and X_1_ had the greatest effect on the particle size of Til NCs.

The optimal conditions for preparing Til NCs by the antisolvent precipitation ultrasonic method were as follows: The optimal NCs were obtained under an ultrasonic power of 129.63 W with a PVA concentration of 0.30% (*w*/*v*) and a TPGS concentration of 0.08% (*w*/*v*) as stabilizers. The particle size of the Til NCs was 94.27 ± 1.19 nm, which was close to the predicted value (85.74 nm). This indicated that the model could be used to fit and analyze the extraction process with a good optimization effect and stable and reliable results.

### 2.2. Characterization of Til NCs

#### 2.2.1. Particle Size and Morphology

The particle size and PDI of Til NCs were determined by the dynamic light scattering method. As shown in Figure 4A,B, the particle size of the fresh Til NCs was 94.27 ± 1.19 nm and the PDI was 0.25 ± 0.04. The PDI value was in the range of 0.3 to 0.5, indicating an acceptable size distribution of the nanocrystals [38]. The zeta potential is very important for nanoparticle colloid stability, because it represents an electrostatic barrier to prevent aggregation and agglomeration of nanoparticles [39]. In this study, the zeta potential for Til NCs was 10.30 ± 0.99 mV, suggesting a positive repulsive effect between nanoparticles and keeping Til NCs stable [40,41]. In addition, Til NCs showed light blue opalescence (Figure 4D, left). When illuminated with a laser, the Til NCs showed a pronounced Tyndall effect, indicating that it was a colloidal system (Figure 4D, right). The morphology of Til NCs was observed by transmission electron microscopy. From the image (Figure 4C left), the crude Til presented micron-sized rod-like particles, while Til NCs presented needle-like nanoparticles (Figure 4C right).

#### 2.2.2. Crystalline Characterization

The X-ray [42] powder diffraction analysis of PVA, TPGS, crude Til, and Til NCs is shown in Figure 4E. The main characteristic peaks (2θ = 11.97°, 12.82°, and 26.00°) were found in the diffraction patterns of the crude Til, indicating the crystalline form of Til. The patterns of PVA and TPGS were not supposed to interfere with crude Til. Til NCs showed no obvious diffraction peak, indicating that Til NCs were in an amorphous state.

The DSC result profiles of PVA, TPGS, crude Til, and Til NCs are shown in Figure 4F. PVA’s DSC curve revealed a subtle endothermic peak, which may have resulted from the material’s partial or progressive crystallization or phase transition process during heating. TPGS revealed a melting peak at 39 °C. The DSC curve of crude Til showed two distinct endothermic peaks at 166 °C and 215 °C, as well as an exothermic peak at 204 °C. It indicated that crude Til had a melting point at 166 °C for one crystal form. As the temperature increased, the crystal form of Til transformed to a more stable state at 204 °C. This stable crystal form exhibited a melting point at 215 °C. Furthermore, Til degradation happened as the temperature increased over 225 °C. For Til NCs, the DSC curve started to exhibit a fairly noticeable endothermal peak at roughly 100 °C, which might be because the water in the Til NCs solution system gradually evaporated during the heating process. Interestingly, no endothermic peak was observed around the equivalent Til melting peak, suggesting that Til NCs were mostly amorphous. This observation aligns with the XRPD analysis.

### 2.3. Stability and Saturation Solubility

As shown in Figure 5A,B, the particle size of Til NCs did not change significantly within 60 d at 25 °C and 4 °C, proving that Til NCs had a good stability.

The saturation solubilities of crude Til and Til NCs are shown in Figure 5C. The saturation solubility of crude Til in water, SGF, and SIF was 7.20, 8.80, and 10.82 μg/mL at 37 °C, respectively. After crude Til was nanosized, the solubility of Til was significantly improved. Quantitatively, the solubility of Til NCs in water, SGF and SIF was 112.36, 225.31, and 433.47 μg/mL, respectively. In addition, the saturation solubility of Til NCs in SIF was approximately twice that of SGF. This may be because the flavonoid structure of its polyphenolic hydroxyl group is more conducive to dissolution in more alkaline conditions.

### 2.4. In Vitro Release of Til NCs

The release behavior of Til NCs was studied by dialysis. As shown in Figure 5D, the dissolution rate of Til NCs was higher than that of crude Til in all three media. In water, the dissolution rate of crude Til and Til NCs was (1.67 ± 0.10)% and (5.59 ± 0.02)%, respectively. In SGF and SIF media (Figure 5E,F), the dissolution rate of Til NCs was higher than that in the water media, possibly resulting from the significant differences in composition and properties between water and SGF or SIF. SGF and SIF often contain hydrochloric acid/phosphoric acid and surfactants to mimic the complex environment of the gastrointestinal tract, which may contribute to the dissolution or dissociation of Til-NCs, thereby facilitating drug release [43]. Meanwhile, the dissolution rate of Til NCs in SGF and SIF was about 20-and 22-fold higher than that of crude Til, respectively. The reasons that Til NCs could significantly improve the solubility of Til were as follows: (1) the smaller particle size of Til NCs resulted in a large surface area and short diffusion distance according to the Noyes–Whitney equation, which facilitated the rapid movement of surface Til molecules into the medium, and (2) the amorphous feature of Til NCs allowed Til molecules to easily escape from the particle without breaking lattice energy.

### 2.5. The Cellular Uptake and Cytotoxicity of Til NCs

The cellular uptake of Til NCs was investigated in RAW264.7 cells. The RAW264.7 cell line is a murine macrophage cell line derived from tumor tissue induced by mouse leukemia virus. It is widely used in the study of inflammation, immune response, and infection because of its typical macrophage properties and ability to mimic the function and response of macrophages in the body. In our study, we selected the RAW264.7 cell line for macrophage polarization investigation [44,45,46]. Confocal laser scanning microscopy (CLSM) was used to investigate the cellular uptake of Til NCs/Cou 6 in RAW264.7 cells. As indicated by the fluorescence images (Figure 6D,E), the uptake of Til NCs/Cou 6 was time dependent. To more accurately quantify the uptake of Til NCs/Cou 6, FCM was used (Figure 6F,G). The results were consistent with those from CLSM.

Subsequently, we investigated the cytotoxicity of crude Til and Til NCs on RAW264.7 cells. Figure 6A,B shows that the cell viability was over 80% after incubation with crude Til and Til NCs for 24 h and 48 h, even at a high concentration of 80 μg/mL. This suggested that Til NCs were safe for RAW264.7 cells.

Previous studies have shown that elevated levels of LPS and IFN-γ are major risk factors for the development of chronic inflammatory diseases, which could induce cell death. To evaluate the protective ability of Til NCs, the viability of RAW264.7 cells stimulated with LPS and IFN-γ was tested after treatment with Til NCs. As shown in Figure 6C, for the control group, the induction of LPS and IFN-γ reduced the viability of RAW264.7 cells. Cell viability increased in a dose-dependent manner with increasing Til concentration, although there was no significant difference between crude Til and Til NCs. This indicates that Til has a cytoprotective ability in anti-inflammatory treatment.

### 2.6. Study of Macrophage Polarization

In this study, we analyzed the effect of Til NCs on LPS- and IFN-γ-induced macrophage polarization by analyzing the proportions of M1 macrophages and M2 macrophages and the M1/M2 ratio after treatment with Til NCs by FCM. A schematic diagram is shown in Figure 7A, and the results are shown in Figure 7B,C. The percentages of CD86- and CD206- positive cells in the model group were 41.20 ± 1.24% and 11.04 ± 2.10%, respectively. These results suggested that LPS and IFN-γ could promote the polarization of macrophages to the M1 type. After treatment with crude Til and Til NCs, although there was no significant difference in CD86-labeled M1 macrophages (Figure 7D), the mean fluorescence intensity of CD206, a marker of M2 macrophages, was significantly increased (*p* < 0.05, *p* < 0.001) (Figure 7E). The mean fluorescence intensity (MFI) of CD206 for Til NCs was approximately 2-fold higher than that of crude Til, indicating that Til NCs showed a superior ability to promote macrophages from the M1 type to the M2 type. Similarly, in the overall M1 (F4/80^+^CD86^+^)/M2 (F4/80^+^CD206^+^) numerical plot (Figure 7F), the crude Til group and Til NCs group sequentially decreased the M1/M2 ratio compared to the model group, indicating the progressive dominance of anti-inflammatory M2 in this polarized system. Taken together, Til treatment selectively promoted M2-type macrophages, with Til NCs exhibiting better effects.

### 2.7. Anti-ROS and Anti-Inflammatory Factor Study

Overproduction of ROS is frequently linked to M1-type macrophage activation and activity. Thus, ROS generation can be enhanced by LPS- and IFN-γ-generated M1-type macrophages [47,48,49]. To investigate the ROS level in RAW264.7 cells treated with LPS and IFN-γ, the dichlorodihydrofluorescein diacetate (DCFH-DA) method was used, and the data were analyzed with Flow Jo software 10.8.1. As shown in Figure 8A, the model group showed high bright green fluorescence, indicating a high level of ROS. Compared with the model group, the fluorescence intensity decreased with the treatment of crude Til, and a negligible fluorescence signal was found when Til NCs were added. Quantitative analysis of the fluorescence density (Figure 8B) showed that Til NCs significantly (*p* < 0.01) reduced ROS levels in inflammatory macrophages compared with crude Til, which meant that ROS were more scavenged by Til NCs. In addition to detecting ROS generation in inflammatory macrophages after Til NCs treatment using CLSM, we used FCM for further experimental verification. As shown in Figure 8C,D, LPS- and IFN-γ-induced macrophages exhibited significantly increased intracellular ROS production and the MFI of ROS compared with the negative control group (*p* < 0.001). The results were similar to the CLSM results, where ROS levels were significantly reduced after Til NCs’ treatment.

To further investigate whether Til NCs inhibited LPS- and IFN-γ-induced macrophage secretion of more proinflammatory cytokines, we analyzed the supernatants of treated macrophages by ELISA (Figure 8E). The expression of TNF-α, IL-1β, IL-6, and IL-10 was tested. Compared with the control group, LPS and IFN-γ (model) significantly increased the levels of TNF-α (539.23 pg/mL), IL-6 (190.93 pg/mL), and IL-1β (397.50 pg/mL) and significantly decreased IL-10 (93.03 pg/mL). Next, we treated LPS- and IFN-γ-induced RAW 264.7 cells with crude Til and Til NCs. As shown, these induced proinflammatory cytokines were significantly inhibited by Til NCs compared to the crude Til group. Specifically, Til NCs significantly (*p* < 0.01) inhibited TNF-α (421.50 pg/mL), IL-1β (271.70 pg/mL), and IL-6 (124.70 pg/mL). In addition, we further investigated the IL-10 level in M1 macrophages. As expected, Til NCs can significantly (*p* < 0.01) increase the expression of IL-10 (135.70 pg/mL) in induced macrophages, even more than crude Til, which may be attributed to the high dissolution rate and saturation effect of nanocrystals, as well as the cellular uptake with intact nanoparticles. Thus, Til NCs are a potent strategy for selectively modulating macrophage polarization and reducing the overproduction of ROS and proinflammatory cytokines.

## 3. Materials and Methods

### 3.1. Materials

Til (purity > 98%) was procured from the State Key Laboratory of Biotherapy, Sichuan University. The Til standard substance (purity > 98%) was purchased from Chengdu Yirui Biotechnology Co., Ltd. (Chengdu, China). Sodium dodecyl sulfate (SDS) and sodium deoxycholate were purchased from Tianjin Yongda Chemical Reagent Co., Ltd. (Tianjin, China). TPGS was purchased from Sahn Chemical Technology Co., Ltd. (Shanghai, China). Poloxamer 188 (F68) and Poloxamer 407 (F127) were obtained from BASF (China) Co., Ltd., (Shanghai, China). PVA was purchased from Japan Kuraray Co., Ltd., Osaka, Japan. Polyvinylpyrrolidone K30 (PVP K30) was purchased from Tianjin Bodi Chemical Co., Ltd. (Tianjin, China). Cetyltrimethylammonium bromide (CTAB) was purchased from Chengdu Rongze Local Chemical Co., Ltd. (Chengdu, China). N,N-Dimethylformamide (DMF) was obtained from Shandong Jinmei Riyue Chemical Co., Ltd. (Qingdao, China). Simulated gastric fluid (SGF) and simulated intestinal fluid (SIF) (LEAGENE^®^, CZ0200). Cell Counting Kit-8 (CCK-8) was purchased from DOJINDO Molecular Technologies. Coumarin 6 (Cou 6) was obtained from J&K Scientific Co., Ltd. (Beijing, China). Dialysis bags (molecular weight cutoff 3500) were purchased from Mbra-Cel^®^, Viskase, Lombard, IL, USA. Chromatographic grade formic acid, methanol, absolute ethanol, acetone, and acetonitrile were obtained from Tianjin Union Science and Technology Co., Ltd. (Tianjin, China). All reagents and chemicals used were of analytical or chromatographic grade.

### 3.2. Methods

#### 3.2.1. Preparation of Til NCs

Til NCs were prepared using an antisolvent precipitation ultrasonication method [[31],[32]，[33]]. In brief, crude Til was completely dissolved in DMF:ethanol (1:1). The Til solution was rapidly injected into an aqueous solution containing different stabilizers (F68, F127, PVP K30, PVA, sodium deoxycholate, SDS, TPGS, or CTAB). The mixture was subjected to ultrasonication at 130 W for 25 min in an ice bath. Finally, a transparent blue opalescent colloidal solution was obtained. Ultimately, dialysis was used to remove the organic solvent.

The Cou 6 labeled Til NCs (Til NCs/Cou 6) were obtained using a similar method [50,51]. Til and an appropriate amount of Cou 6 were mixed in a DMF and ethanol (1:1, *v*/*v*) solvent. The remaining procedures were the same as those in the preparation of the Til NCs method described above. The original Til NCs/Cou 6 suspension was centrifugated at 15,000 rpm for 15 min and resuspended with deionized water to remove the residual Cou 6. After three centrifugation–resuspension cycles, the Til NCs/Cou 6 samples were eventually acquired.

#### 3.2.2. Formulation Optimization of Til NCs

The preparation factors, such as stabilizer type, stabilizer content, ultrasound power, and ultrasound time, were screened using the single-factor technique (Table 3). Three important variables were found in this study: TPGS concentration, PVA concentration, and ultrasound power. Subsequently, Til nanoparticle formulation and process parameters were further optimized using CCD utilizing Design-Expert software (Version 12, Stat-Ease Inc., Minneapolis, MN). CCD is a statistical technique that fits the functional relationship between each factor and the response value using the multiple quadratic regression equation. The mathematical model’s variance analysis is then used to determine the ideal process parameters. The three primary variables (X_1_ being ultrasonic power, X_2_ being PVA concentration, and X_3_ being TPGS concentration) were identified as the critical process parameters influencing Til NCs’ particle size (Y). The factors and their levels are listed in Table 4. To lessen systematic errors brought on by outside influences, the studies were conducted at random.

### 3.3. Characterization of Til NCs

The particle size distribution and zeta potential of Til NCs were determined using a Malvern Zeta Nanosizer (Zetasizer Nano ZS90; Malvern Instruments Ltd., Malvern, UK). Each sample was measured in triplicate at room temperature. The morphology was assessed using transmission electron microscopy (TEM) (JEM-1400PLUS, JEOL Ltd., Tokyo, Japan) after negative staining. The crystal form of Til NCs was analyzed by powder X-ray diffraction (PXRD) (Rigaku Ultima IV, Osaka, Japan) and differential scanning calorimetry (DSC) (TA Q20, METTLER TOLEDO, Viskase, Lombard, IL, USA).

### 3.4. Stability and Saturation Solubility of Til NCs

The particle size and PDI of Til NCs were tracked while the storage stability of the material was investigated at two different temperatures (25 °C and 4 °C). In the meantime, by introducing an excess of Til NCs, the saturated solubility of Til NCs in water, SGF, and SIF was examined. The samples underwent a 24 h shake-test at 37 °C, a 10 min centrifugation at 13,000 rpm, and a filtering process via 0.45 μm microporous membranes before the drug concentrations were ascertained using HPLC.

### 3.5. In Vitro Release

Using the dialysis diffusion method, the in vitro drug release behavior of Til NCs in various mediums at 37 °C was ascertained. In a shaker (Julabo, SW 23, Seelbach, Germany), Til NCs and crude Til (1 mg/mL, 1 mL) were placed in individual dialysis bags (molecular weight cutoff 3500) and exposed to 50 mL of water, SGF (pH 1.2, containing 0.5% (*v*/*v*) Tween 80, 0.32% (*w*/*v*) pepsin), or SIF (pH 6.8, 0.5% (*v*/*v*) Tween 80, 1% (*w*/*v*) trypsin) at 37 °C and gently shaken (100 rpm). One milliliter of the external liquid was taken out at predetermined intervals and replaced with the same volume of brand-new medium (37 °C). Samples were centrifuged at 12,000 rpm for 10 min, and the Til concentration was determined using HPLC.

### 3.6. Cell Culture

The Cell Resource Bank of the Chinese Academy of Sciences (Shanghai, China) provided the RAW264.7 cell line. High-glucose DMEM medium (Invitrogen, San Diego, CA, USA) was supplemented with 10% fetal bovine serum (FBS; Gibco, San Diego, CA, USA) and 1% penicillin and streptomycin (HyClone, San Diego, CA, USA).

### 3.7. Cellular Uptake

To assess the cellular uptake of the Til NCs. We used CLSM (LSM880, Carl Zeiss, Heidelberg, Germany) to further investigate the efficiency of Til NCs’ uptake by RAW264.7 cells. In brief, RAW264.7 cells were seeded in a confocal well (size 15 mm) at a density of 1 × 10^4^ cells/well and incubated for 24 h. After stimulation with LPS and IFN-γ, 100 μL of Til NCs/Cou 6 was added and incubated for different times. The cells were then washed, fixed with fixation buffer (BioLegend^®^, San Diego, CA, USA, B382280), and stained with DAPI. The cells were then rinsed twice with PBS and visualized under CLSM. Moreover, RAW264.7 cells were seeded onto 24-well plates (1 × 10^5^ cells/well) and incubated overnight.

To evaluate the Til NCs’ cellular uptake, we employed CLSM (LSM880, Carl Zeiss, Germany) to look at the effectiveness of RAW264.7 cells’ uptake of Til NCs. To sum up, 1 × 10^4^ cells/confocal well (size 15 mm) was seeded with RAW264.7 cells, and the well was incubated for 24 h. Following the application of LPS and IFN-γ activation, 100 μL of Til NCs/Cou 6 was introduced and incubated at varying periods. The cells were subsequently stained with DAPI, fixed with fixation buffer (BioLegend^®^, B382280), and rinsed. After two PBS rinses, the cells were examined under CLSM. Further, 1 × 10^5^ RAW264.7 cells were planted into each well of a 24-well plate, and the plates were left overnight to incubate. For a whole day, LPS (100 ng/mL) and IFN-γ (20 ng/mL) were used to activate RAW264.7 cells. Next, 100 μL of Til NCs/Cou 6 (100 ng/mL of Cou 6) was introduced and the mixture was co-cultured at 37 °C. After that, PBS was used twice to wash the cells. Then, Til NC uptake was measured using FCM (NovoCyteTM, ACEA Biosciences, Inc., Beijing, China).

### 3.8. Cytotoxicity Analysis

RAW264.7 cells were applied to assess the toxicity of the Til NCs using a CCK-8 assay. The cells were seeded in 96-well plates at a density of 1 × 10^4^ cells/well and cultured for 24 h in a 5% CO_2_ incubator (HERAcell 150, Thermo Fisher Scientific, Inc., Waltham, MA, USA). After removing the culture medium, the cells were incubated with Til NCs and crude Til. Then, after incubation for 24 h and 48 h, the samples were tested using the CCK-8 kit reagent. In addition, LPS and IFN-γ are the two major triggers of macrophage inflammation, which not only induce the formation of inflammatory cells but also reduce cell activity. Therefore, the protective ability of Til NCs against LPS and IFN-γ was also evaluated. CCK-8 was used to detect the effect on cell viability after drug induction. The absorbance at 450 nm was then used to calculate the optical density (OD) values using a microplate reader (Biotek, Winooski, VT, USA). There were six duplicate analyses of each sample.

### 3.9. Macrophage Polarization Analysis

M1/M2 polarization of RAW264.7 cells was detected by FCM. In short, the cells were seeded in 24-well plates at a density of 1 × 10^5^ cells/mL in triplicate. The cells were stimulated with LPS and IFN-γ, the old medium was discarded, and different groups of drugs were added (Til, 20 μg/mL). After 12 h of incubation, the cells were gently aspirated into a flow tube, washed with PBS, and sequentially stained for 30 min at 4 °C in the dark with FITC-labeled anti-murine F4/80 antibody, PE-labeled anti-murine CD86, and APC-labeled anti-murine CD206. After that, the cells were washed, resuspended, and detected by FCM. The results were analyzed with Flow Jo software (Tree star, Chico, CA, USA).

### 3.10. Assessment of Intracellular ROS Generation

Following treatment, cells were treated for 30 min with DCFH-DA. The DCFH-DA test was used to measure the levels of intracellular ROS, as previously reported [52,53]. Finally, picture acquisition was accomplished using CLSM. By using FCM at a wavelength of 488 nm, the proportion of ROS-positive cells was ascertained and analyzed using Flow Jo software.

### 3.11. Cytokine Assays

RAW264.7 cells were cultured in 24-well plates for 24 h at a density of 1 × 10^5^ cells/mL. After receiving a 24 h Til NCs therapy, the cells were stimulated with LPS and IFN-γ for a further 24 h. The concentrations of TNF-α, IL-6, IL-1β, and IL-10 in the cell supernatant were measured using ELISA kits.

### 3.12. Analytical Methods

#### 3.12.1. HPLC Analysis

A Waters e2695 HPLC system (Waters Technology Co., Ltd., Shanghai, China) was used to calculate the amount of Til. Til was detected at a wavelength of 323 nm and separated on a Reliasil C18 column (4.6 mm × 250 mm, 5 μm). Acetonitrile and 0.5% formic acid (25:75) made up the mobile phase, and the temperature of the column was 40 °C. The flow rate was 1.0 mL/min [3].

#### 3.12.2. Statistical Analysis

Every quantitative piece of information was presented as the average ± standard deviation (SD) of three or more measurements. When comparing two groups, unpaired *t*-tests were employed, and when comparing several groups, a one-way analysis of variance was employed: * *p* < 0.05, ** *p* < 0.01, *** *p* < 0.001, and **** *p* < 0.0001.

## 4. Conclusions

In summary, our study has achieved significant success in preparing Til NCs via the combination of antisolvent precipitation and ultrasonication techniques. Specifically, the optimal formulation and preparation process of Til NCs were optimized by CCD on the basis of the single-factor method regarding of the parameters of organic solvent type, oil-to-water ratio, ultrasonication power and time, and stabilizers type. As a result, the optimal Til NCs was obtained by using a mixed organic solvent of DMF and ethanol (1:1, *v*/*v*) with an oil-to-water ratio of 1:20. The NCs were stabilized by PVA (0.30%, *w*/*v*) and TPGS (0.08%, *w*/*v*). The ultrasonic power and time were 130 W and 25 min. In addition, the optimal Til NCs were in amorphous needle-like nanoparticle with a remarkably small particle size (94.27 ± 1.19 nm) and a positive charge (10.30 ± 0.99 mV). Notably, Til NCs exhibited a significantly enhanced water solubility and in vitro dissolution rate compared to crude Til. This enhancement not only improved the bioavailability of Til but also potentiated its anti-inflammatory properties. Specifically, Til NCs demonstrated a robust ability to promote the transformation of macrophages from the M1 to M2 phenotype by increasing cellular uptake, inhibiting ROS and proinflammatory cytokines, and promoting the production of anti-inflammatory cytokines. In conclusion, our research underscores the potential of amorphous Til NCs as a promising therapeutic strategy for the treatment of inflammatory diseases.

## Figures and Tables

**Figure 1 pharmaceuticals-17-00654-f001:**
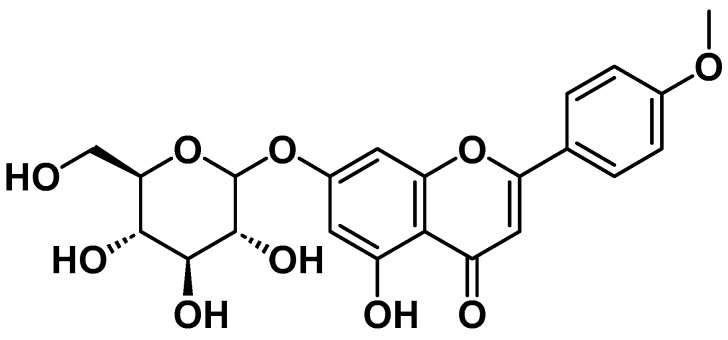
Chemical structure of Til.

**Figure 2 pharmaceuticals-17-00654-f002:**
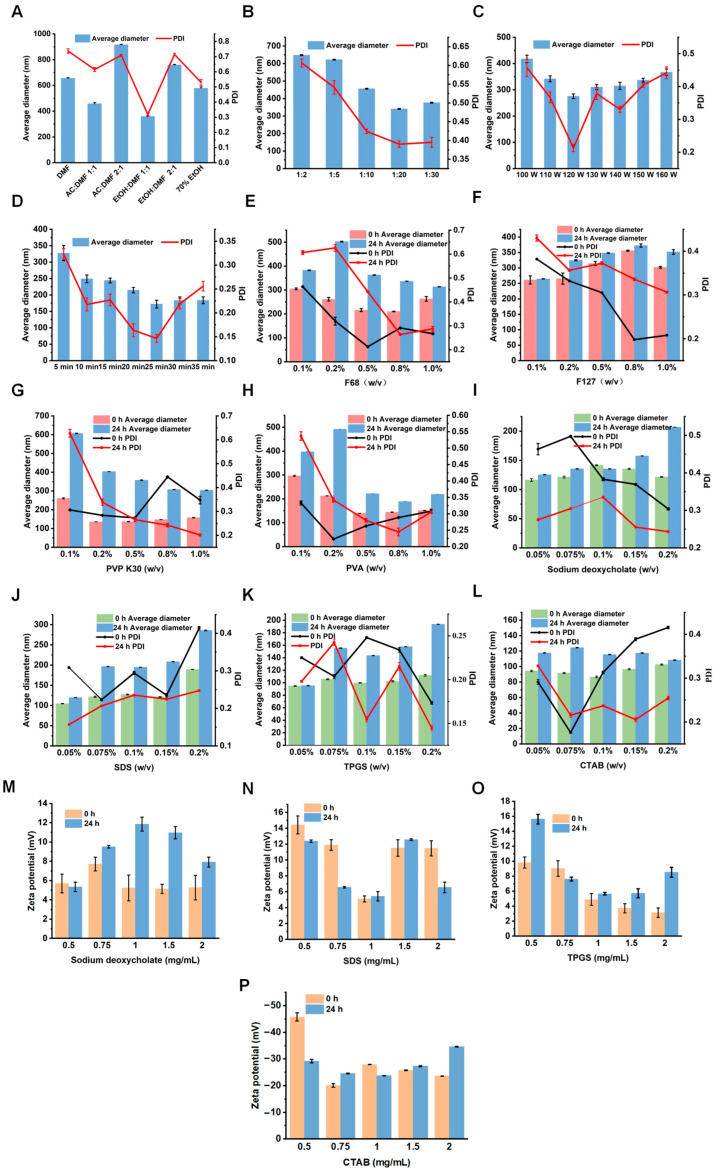
The effect of various parameters on particle sizes and PDIs of Til NCs: (**A**) types of organic solvent, (**B**) ratios of organic phase to water phase, (**C**) ultrasonic power, and (**D**) ultrasonic time. (**E**–**H**) Particle sizes and PDIs of Til NCs with different spatial stabilizers (F68, F127, PVP K30, and PVA) for 0 h and 24 h after preparation. (**I**–**L**) Particle sizes, PDIs and (**M**–**P**) zeta potential of Til NCs stabilized with different ionic stabilizers (sodium deoxycholate, SDS, TPGS, and CTAB) (*n* = 3, mean ± S.D.).

**Figure 3 pharmaceuticals-17-00654-f003:**
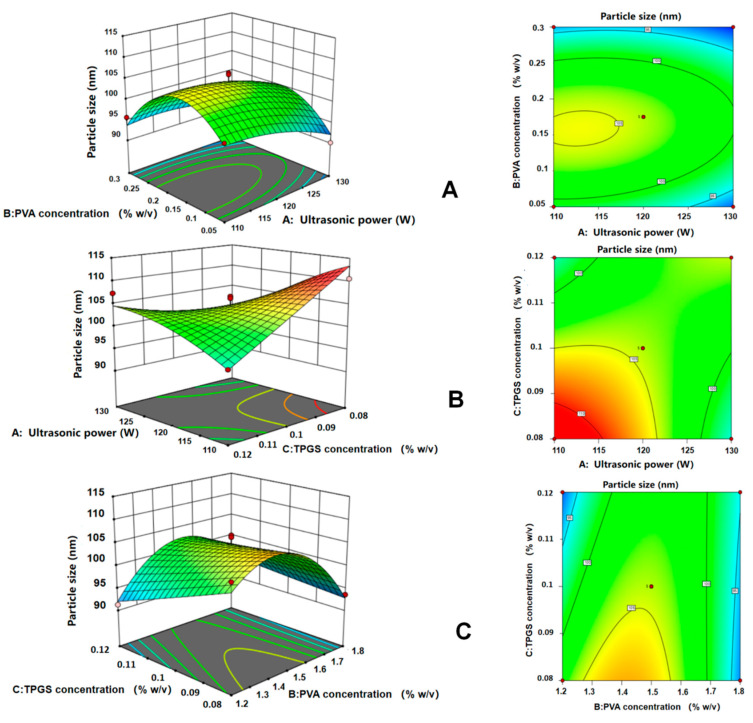
Response-surface analysis shows the effect of ultrasonic power (X_1_), PVA concentration (X_2_), and TPGS concentration (X_3_) on the particle size (Y) of Til NCs. (**A**) Effect of the second-order interaction between ultrasonic power and PVA concentration (X_1_X_2_) on the particle size of Til NCs. (**B**) Effect of the second-order interaction between ultrasonic power and TPGS concentration (X_1_X_3_) on the particle size of Til NCs. (**C**) Effect of the second-order interaction between PVA concentration and TPGS concentration (X_2_X_3_) on the particle size of Til NCs.

**Figure 4 pharmaceuticals-17-00654-f004:**
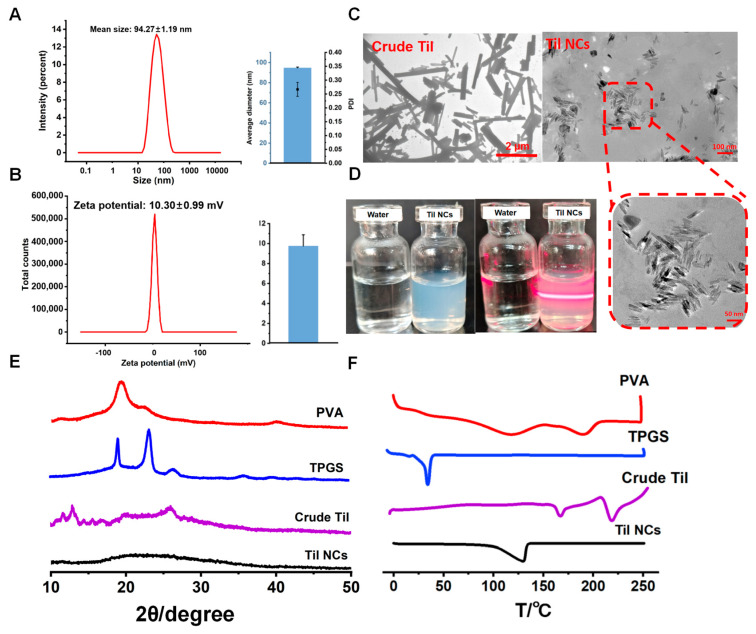
Characterization of Til NCs. (**A**,**B**) The particle size and zeta potential of postoptimized Til NCs. (**C**) TEM images of crude Til and Til NCs (left. Bar = 2 µm, right. Bar = 100 nm). (**D**) The appearance of Til NCs dispersed in water with (right) or without (left) laser irradiation. (**E**) The XRPD patterns for PVA, TPGS, crude Til, and Til NCs. (**F**) DSC curves of PVA, TPGS, crude Til, and Til NCs.

**Figure 5 pharmaceuticals-17-00654-f005:**
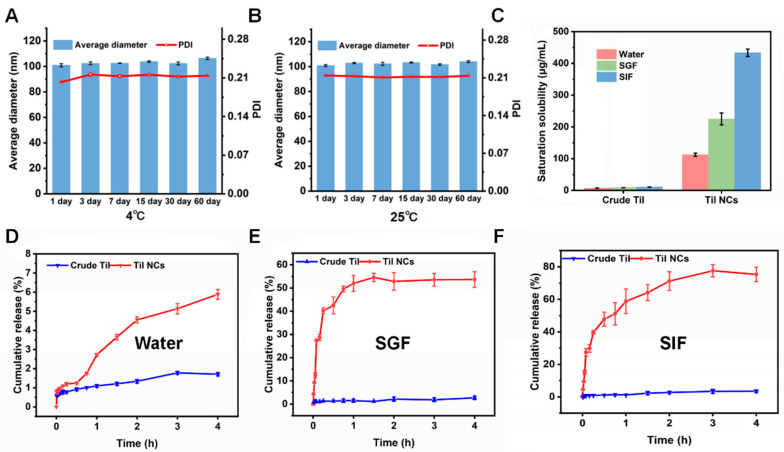
(**A**,**B**) Stability study of Til NCs at (**A**) 4 °C and (**B**) 25 °C. The particle size and PDI of Til NCs were monitored for 60 days. (**C**) Saturation solubility and (**D**–**F**) cumulative release of crude Til and Til NCs in water, SGF, and SIF (*n* = 3, mean ± SD).

**Figure 6 pharmaceuticals-17-00654-f006:**
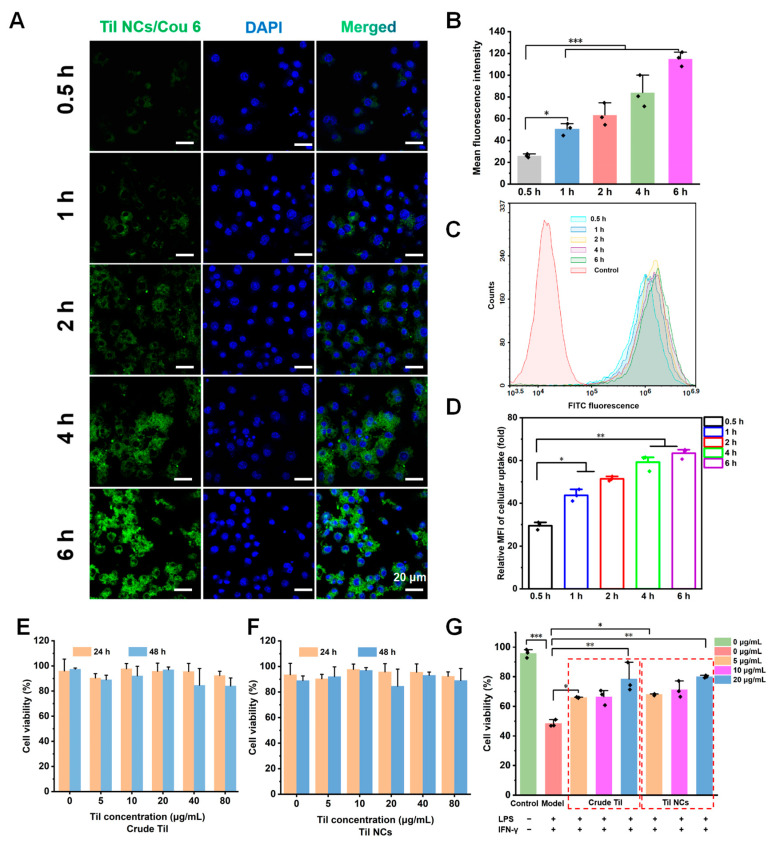
Time-dependent cellular uptake of Til NCs/Cou 6 by RAW264.7 cells. The cells were incubated with Til NCs for different periods (0.5 h, 1 h, 2 h, 4 h, and 6 h) and evaluated with CLSM and FCM. (**A**) Fluorescence images of CLSM with nuclei stained with DAPI (blue). (**B**) Quantitative fluorescence intensity data for CLSM. (**C**) Representative FCM graphs and (**D**) quantified MFI data of FCM. Cytotoxicity and cytoprotective evaluation of crude Til and Til NCs. (**E**,**F**) Cell viability of RAW264.7 cells after incubation with various concentrations of crude Til and Til NCs for 24 h and 48 h. (**G**) Cell viability of RAW 264.7 cells co-treated with LPS (100 ng/mL) and IFN-γ (20 ng/mL) and Til preparation at different concentrations from 5 to 20 μg/mL (the scale bar is 20 µm. *n* = 3, * *p* < 0.05, ** *p* < 0.01, and *** *p* < 0.001).

**Figure 7 pharmaceuticals-17-00654-f007:**
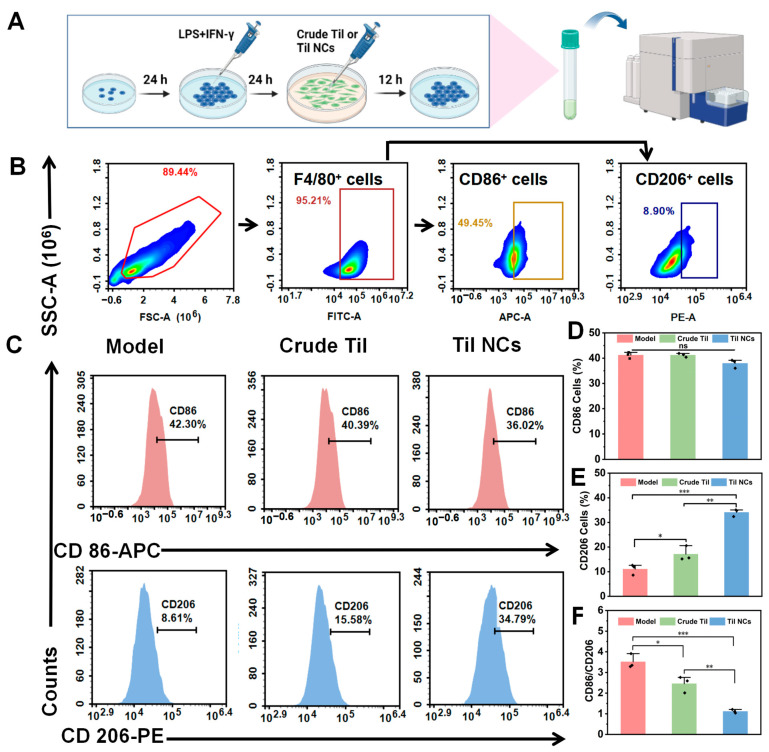
Polarization study of macrophages treated with crude Til and Til NCs in vitro. (**A**) Schematic representation of the macrophage polarization model. (**B**,**C**) FCM was used to investigate the effect of crude Til and Til NCs on the polarization of stimulated RAW264.7 cells. CD86 and CD206 were used as biomarkers of M1 macrophages and M2 macrophages, respectively. (**D**–**F**) Statistical analysis of the expression of CD86, CD206, and M1/M2 after treatment with crude Til and Til NCs. The results are expressed as the means ± SD. *n* = 3, A one-way ANOVA test of multiple comparisons followed by Dunnett’s post hoc test was used in all analyses (* *p* < 0.05, ** *p* < 0.01, *** *p* < 0.001, *ns*, not significant).

**Figure 8 pharmaceuticals-17-00654-f008:**
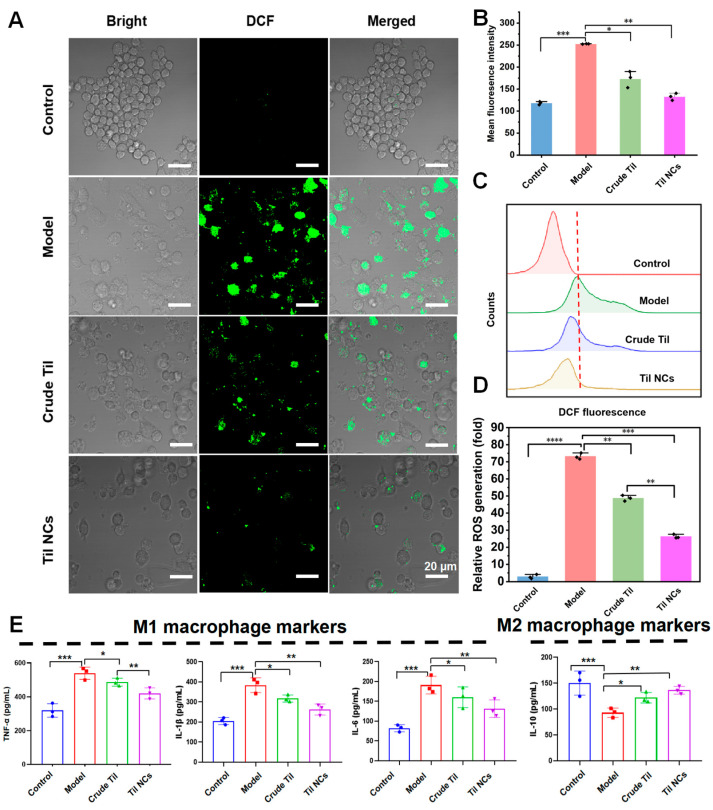
The ability of crude Til and Til NCs to inhibit ROS and inflammatory factor generation in induced RAW264.7 cells. (**A**,**B**) Representative fluorescence images showing intracellular ROS generation in RAW264.7 cells treated with various formulations. ROS were stained with DCF-DA (green). Scale bar = 20 µm. (**C**,**D**) Representative FCM profiles and quantitative results of intracellular ROS generation after different treatments. (**E**) The effects of crude Til and Til NCs treatment on the expression of TNF-α, IL-1β, IL-6, and IL-10. Data are expressed as the mean ± S.D. *n* = 3, A one-way ANOVA test of multiple comparisons followed by Dunnett’s post hoc test was used in all analyses (* *p* < 0.05, ** *p* < 0.01, *** *p* < 0.001, **** *p* < 0.0001).

**Table 1 pharmaceuticals-17-00654-t001:** Arrangements and response variables of CCD.

No.	Argument		Response Value	
	X_1_ (W)	X_2_ (%, *w*/*v*)	X_3_ (%, *w*/*v*)	Y (nm)
1	120	0.3	0.12	93.2
2	120	0.175	0.1	103.4
3	130	0.05	0.1	90.1
4	120	0.3	0.08	93.6
5	130	0.175	0.12	107.3
6	120	0.175	0.1	101.7
7	120	0.175	0.1	106.7
8	120	0.175	0.1	103.8
9	120	0.05	0.12	91.3
10	110	0.175	0.08	110.6
11	120	0.175	0.1	106.4
12	110	0.3	0.1	95.7
13	110	0.05	0.1	99.4
14	130	0.3	0.1	90.1
15	120	0.05	0.08	104.6
16	110	0.175	0.12	96.9
17	130	0.175	0.08	95.9

**Table 2 pharmaceuticals-17-00654-t002:** ANOVA in the quadratic model for responses (Y).

Source	Particle Size (nm)			
	Sum of Squares	df	F-Value	*p*-Value (prob > F)
Model	647.78	9	9.79	0.0033
X_1_	46.08	1	6.27	0.0408
X_2_	20.48	1	2.78	0.1391
X_3_	32.00	1	4.35	0.0754
X_1_X_2_	3.42	1	0.4653	0.5171
X_1_X_3_	157.50	1	21.42	0.0024
X_2_X_3_	41.60	1	5.66	0.0490
X_1_^2^	13.30	1	1.81	0.2206
X_2_^2^	324.40	1	44.11	0.0003
X_3_^2^	0.0221	1	0.0030	0.9578
Residual	51.48	7		
Lack of Fit	33.93	3	2.58	0.1912
Pure Error	17.55	4		
Cor Total	699.26	16		

**Table 3 pharmaceuticals-17-00654-t003:** Single-factor test table of Til NCs.

Factor	Value
Organic solvent	DMF, AC:DMF (1:1, 2:1), EtOH:DMF (1:1, 2:1), 70% EtOH
Oil–water phase ratio	1:2, 1:5, 1:10, 1:20, 1:30
Ultrasonic power (W)	100, 110, 120, 130, 140, 150, 160
Ultrasonic time (min)	5, 10, 15, 20, 25, 30, 35
Spatial stabilizer	PVP K30, PVA, F68, F127
Content of spatial stabilizer (%, *w*/*v*)	0.1, 0.2, 0.3, 0.4, 0.5
Ion stabilizer	sodium deoxycholate, SDS, TPGS, CTAB
Content of ion stabilizer (%, *w*/*v*)	0.050, 0.075, 0.100, 0.150, 0.200

**Table 4 pharmaceuticals-17-00654-t004:** Independent factors and responses in the CCD.

Element	Level		
	Low (−1)	Middle (0)	High (+1)
X_1_, Ultrasonic power (W)	110	120	130
X_2_, PVA concentration (%, *w*/*v*)	0.05	0.175	0.3
X_3_, TPGS concentration (%, *w*/*v*)	0.08	0.1	0.12
Dependent Variables	Goal		
Y, Particle size (nm)	Minimize		

## Data Availability

The data are contained within the article.

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
