# Peer review of "Enhanced Anti-Inflammatory Activity of Tilianin Based on the Novel Amorphous Nanocrystals"

_pharmaceuticals, 2024, doi:10.3390/ph17050654_

Round 1

Reviewer 1 Report

Comments and Suggestions for Authors

 Enhanced Anti-Inflammatory Activity of Tilianin Based on the Novel Amorphous Nanocrystals

Dear Editor,

I have gone through carefully the whole manuscript.

It is well established with authentic research information where I observed the extraction of Til NCs using the anti-solvent precipitation ultrasonic method with a particle size ranging from 90 to 130 nm. However, I have no serious questions to authors. I accept the manuscript without any changes except typos and English mistakes. Rest of it is beautifully narrated and presented the manuscript.

1. Authors should figure caption as Scheme 1 or Fig 1, nothing they have provided.

2. Authors have not provided the NMR details of the Til, since it is very important information for the readers.

3. This study is really helpful to understand the selection of organic solvents, stabilizers and process parameters, however, solvent should be like solvents?

4. The optimal conditions for preparing Til NCs by anti-solvent precipitation ultrasonic method were as follows:

For the above method, need suitable citations.

5. Authors should provide citations for these analysis to compare by the readers.

6. Since, Til NCs showed very nice chromophores in the Fig.4d, but I don’t find any UV-Vis and FT-IR information.

7. The conclusion part should be elaborated as several studies were examined.

Author Response

请参阅附件!

Reviewer 2 Report

Comments and Suggestions for Authors

Dear Author,

This paper is an interesting investigation on the preparation of Tilianin nanocrystals using anti-solvent precipitation ultrasonic method. Furthermore, the optimization of the formulation was carried out with the selection of the parameters influencing the formulation through CCD method.

I suggest to modify the title, or at least, to delete the article “the” from it.

In the abstract, it is important to mention the formulation preparation method.

It is just a curiosity: you used COU-6 in the preparation of Til-NS to evaluate cellular uptake. As COU-6 is not soluble in water and the final formulation the o/w ratio is 1:20, did COU-6 influence in any way Til-NS characteristics?

Page 2, line 41: “acacein-7-glucoside” instead of acacetin-7-glucoside.

Page 3, line 82: “Nanocrystals have unique advantages, such as high drug loading…”. It is not proper to refer to drug loading in the case of nanosuspension because there is no carrier. I suggest substituting it with drug content.

Page 12, figure 4, line 376: Til NCs dispersed in water with laser irradiation are on the right, not on the left.

In Vitro Release of Til-NCs.

How do you explain that the cumulative release of Til-NCs in water is still so low (around 5%) in comparison with the release in SGF, approximately around 50%?

Study of Macrophage Polarization.

In the first part of this paragraph (lines 446 to 465), there is some information about macrophages that must be reported in the introduction. In this part, it is better to report and comment only on the data of the paper.

Therefore, my suggestion is to move this part in the introduction.

Comments on the Quality of English Language

Minor editing of English language required

Author Response

请参阅附件!

Reviewer 3 Report

Comments and Suggestions for Authors

 The manuscript entitled “Enhanced Anti-Inflammatory Activity of Tilianin Based on the 2 Novel Amorphous Nanocrystals” has been reviewed. In general, the study holds interest for the authors. However, some improvements need to be made.

1.      A reference is necessary for lines 79-81.

2.      Line 85: There is insufficient information about the anti-solvent precipitation method. An explanation is necessary.

3.      Line 165: The cell line RAW264.7 is used, but there is no information about its origin or why you chose it. It's necessary to explain.

4.      Figure 2 is too busy. It is necessary to clarify.

5.      Line 400: The solubility of Til NCs in water is around 2 fold lower than that in SGF. However, in Figure 5E, the drug release of Til NCs in SGF is around 10-fold higher. Why does it have such a significant difference?  

6.      In conclusion, it should be explained which formulations and parameters are optimal for this kind of NCs preparation, because there are several parameters and conditions involved. 

Author Response

请参阅附件!

Round 2

Reviewer 3 Report

Comments and Suggestions for Authors

The authors have revised it according to the comments. 

Author Response

Dear reviewer,

First of all, thank you for your review and guidance of my paper. However, I checked the system carefully, but unfortunately I did not see your specific review comments. To ensure that the revisions to the paper fully and accurately reflect your suggestions,

In the future, if you have any questions or suggestions about the revised manuscript, please feel free to contact us. We look forward to your valuable comments and hope that our work can be recognized and supported by you.

Thanks again for your review and guidance!

                                                          Min Sun

                                                         2024.5.12